# Sort-free Gaussian Splatting via Weighted Sum Rendering

Qiqi Hou[1], Randall Rauwendaal[2], Zifeng Li[2], Hoang Le[1], Farzad Farhadzadeh[1], Fatih Porikli[1],
Alexei Bourd[2], Amir Said[*1]

[1]Qualcomm AI Research[†]        [2]Graphics Research Team

[1]{qhou,rrauwend,zifeli,hoanle,ffarhadz,abourd,fporikli,asaid}@qti.qualcomm.com

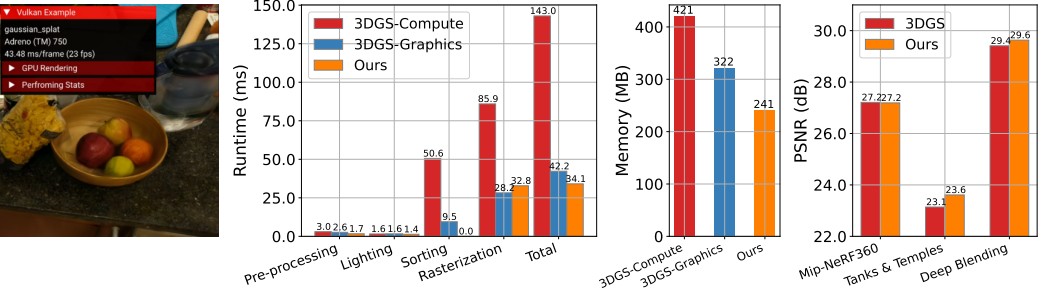

| (a) Run on Mobile Phones | (b) Runtime Comparison | (c) Memory Comparison | (d) PSNR comparison |

Figure 1: This paper proposes a sort-free Gaussian Splatting method, which simplifies volumetric rendering to Weighted Sum Rendering. (a) Visual example from the "counter" scene rendered on a Snapdragon® 8 Gen 3 GPU[‡]. (b) and (c) Runtime and memory comparison between our methods and 3DGS Kerbl et al. (2023) at a resolution of $1920 \times 1080$ on the "counter" scene , respectively. We implemented two versions of 3DGS, one implementation faithfully porting 3DGS CUDA kernels to Vulkan compute shaders (3DGS-Compute), and another leveraging a global-sort following by hardware rasterization (3DGS-Graphics). (d) PSNR results on Mip-NeRF, Tank & Temples, and Deep Blending datasets.

## Abstract

Recently, 3D Gaussian Splatting (3DGS) has emerged as a significant advancement in 3D scene reconstruction, attracting considerable attention due to its ability to recover high-fidelity details while maintaining low complexity. Despite the promising results achieved by 3DGS, its rendering performance is constrained by its dependence on costly non-commutative alpha-blending operations. These operations mandate complex view dependent sorting operations that introduce computational overhead, especially on the resource-constrained platforms such as mobile phones. In this paper, we propose Weighted Sum Rendering, which approximates alpha blending with weighted sums, thereby removing the need for sorting. This simplifies implementation, delivers superior performance, and eliminates the "popping" artifacts caused by sorting. Experimental results show that optimizing a generalized Gaussian splatting formulation to the new differentiable rendering yields competitive image quality. The method was implemented and tested in a mobile device GPU, achieving on average $1.23\times$ faster rendering.

## 1 Introduction

Photo-realistic 3D view synthesis has very wide use in graphics applications like video games, virtual reality, and 3D scene modeling techniques based on learnable appearance and transparency have achieved remarkable success thanks to their capability for consistently generating image details. Neural Radiance Fields (NeRF) by Mildenhall et al. (2021) is one seminal work using this

---

[*]Corresponding author

[†]Qualcomm AI Research is an initiative of Qualcomm Technologies, Inc.

[‡]Snapdragon and Qualcomm branded products are products of Qualcomm Technologies, Inc. and/or its subsidiaries.

approach, where Multi-Layer Perceptron (MLP) networks and positional encoding are used to create models representing a scene's radiance field and opacity.

More recently, 3D Gaussian Splatting (3DGS) was proposed by Kerbl et al. (2023), and rapidly gained popularity (Fei et al., 2024; Chen & Wang, 2024b; Wu et al., 2024). It replaces the MLP networks used by NeRF with a sparse scattering of anisotropic 3D Gaussian "splats" that have view-dependent appearance. These splats are efficiently rendered using rasterization instead of costly volumetric ray-marching, while maintaining its rendering differentiability. Nevertheless, while 3DGS produces high quality views with significantly lower complexity compared to NeRF, it is still challenging for resource-constrained platforms like mobile phones.

An important complexity factor results from the fact that splatting volumetric elements with transparency requires sorting before $\alpha$-blending, and it is difficult to combine non-commutative blending operations—which are intrinsically sequential—with the parallelization required for fast rendering. Furthermore, approximating ray integrals with sorting can create visible "popping" transitions due to temporal changes in rendering order, that can only be alleviated with more complex sorting and rendering (Radl et al., 2024).

The original 3DGS proposal achieves impressive rendering performance with a compute-based rasterizer implemented in CUDA, enabled by a tiled-based GPU radix sort that ensures front-to-back $\alpha$-blending, but incurring memory overhead. Subsequent implementations have leveraged other APIs such as Vulkan or WebGL kishimisu (2024); Kwok (2024), either choosing a similar compute-based implementation, or leveraging the graphics hardware (GPU) to varying degrees. Unfortunately, there is no mechanism to feed the hardware rasterizer in a tile-wise fashion, so such implementations must either perform a costly global-sort prior to submitting work to the hardware rasterizer, or maintain a compute-based rasterizer, sacrificing the benefits of hardware execution. In the case of WebGL, implementations often rely on an asynchronous CPU-sort that runs at a lower-frequency than the frame rate, which degrades view quality and exacerbates temporal "popping" artifacts.

Although the sorting stage in Gaussian Splatting appears inevitable, it results from using approximations to physical processes. In reality, a general scheme for learning scene representations, with components and parameters optimized for the best view reproductions, is bound only by the mathematical constraints of the scene model. This observation motivates us to pursue alternative methods that reduce rendering complexity while incurring minimal degradation in view quality.

We are further encouraged by a widely adopted empirical method for rendering transparent media utilizing the traditional rendering pipeline — Order Independent Transparency (OIT) Meshkin (2007); McGuire & Bavoil (2013), representing a class of techniques in rasterization-based computer graphics for rendering transparency without sorting. By eliminating the sorting step from the rasterization pipeline, OIT maintains rendering speed and has been incorporated into commercial tools such as Blender, Vulkan, Unity, and Unreal Engine.

In this paper we exploit those approaches to develop a novel view synthesis method combining learning with commutative blending operations. It extends the definition of learned Gaussian Splatting, and integrates it with a new method for volumetric rendering, called Weighted-Sum Rendering (GS-WSR), still relying on differentiable rendering and machine learning techniques. To render an image, our approach employs the depth of each Gaussian to compute weights for each Gaussian splat using learned functions. These weighted Gaussians are then splatted onto the image plane using a straightforward summation operation, thereby obviating the necessity for sorting. As a further enhancement, we found that introducing view-dependent opacity greatly improved image quality for sort-free Gaussian Splatting.

This paper contributes to Gaussian Splatting as follows. First, we present the first sort-free Gaussian Splatting technique which is compatible with the graphics pipeline, and allows us the benefit of hardware rasterization. Second, we introduce Weighted Sum Rendering as well as view-dependent opacity that can effectively learn the model parameters and produce high-quality rendered images. Third, our experiments show that our method accelerates Gaussian Splatting with comparable visual performance on mobile phones.

## 2 RELATED WORK

**Neural Scene Representations**. Recent advancements in neural scene representations for novel view synthesis have led to significant progress, with techniques assigning neural features to structures like volumes Lombardi et al. (2019); Sitzmann et al. (2019), textures Chen et al. (2020); Thies et al. (2019), or point clouds Aliev et al. (2020). The pioneering NeRF (Neural Radiance Fields) approach Mildenhall et al. (2021), revolutionized the field by using Multi-Layer Perceptrons (MLPs) to encode 3D density and radiance without proxy geometry, achieving photorealistic renderings from sparse 2D images. Subsequent research has aimed to enhance NeRF's quality and efficiency through optimized sampling strategies Neff et al. (2021), light field-based formulations Li et al. (2022), compact representations Müller et al. (2022), tensor decomposition Chen et al. (2022), or leveraging the polygon rasterization pipeline for efficient rendering on mobile devices such as MobileNeRF Chen et al. (2023). However, the high computational and memory requirement still puts a challenge on mapping these methods on resource constrained platforms.

**Order Dependent Transparency 3DGS** The novel 3D Gaussian Splatting (3DGS) and its numerous follow-up works primarily utilize anisotropic 3D Gaussians for scene representation and an efficient differentiable rasterizer to project these Gaussians onto the image plane, either in pixel or feature representation Hamdi et al. (2024); Lin et al. (2024); Yu et al. (2024); Liang et al. (2024); Yan et al. (2024); Lu et al. (2024); Bulò et al. (2024); Fan et al. (2024); Fu et al. (2024); Zhang et al. (2024); Zou et al. (2024); Jiang et al. (2023); Qin et al. (2024); Li et al. (2024); Jo et al. (2024); Lee et al. (2024); Fan et al. (2023); Niedermayr et al. (2024); Chen et al. (2024); Morgenstern et al. (2023); Navaneet et al. (2023). These methods enable fast, high-resolution rendering while maintaining excellent quality. For a comprehensive overview, please refer to recent survey papers such as Chen & Wang (2024a). A crucial requirement in these methods for proper blending and rendering is the order of the Gaussians, which are typically sorted by their depth using a tile-based sorting algorithm. However, this sorting requirement introduces several challenges in practical implementation and visual quality, such as sudden changes in the appearance of object parts or "popping" artifacts, as recently addressed in Radl et al. (2024). While their results show a limited increase in computational complexity, the necessary sorting modifications are significantly more involved.

**Order Independent Transparency**. Modeling partial coverage has a long history in computer graphics. It's an essential problem as we need to render fine non-opaque structure or elements such as flames, smoke, clouds, hair, etc. In Porter and Duff's seminal work Porter & Duff (1984), they formulate transparency as the **OVER** operator as

$$\mathbf{C} = \alpha_0 \mathbf{c}_0 + (1 - \alpha_0)\mathbf{c}_1, \tag{1}$$

where $\alpha$, $\mathbf{c}$ indicate alpha and color, respectively. As the **OVER** operator is not commutative, it requires back-to-front order for composition. Traditional accelerated either by successively "peeling" depth layers Everitt (2001) or accumulating list for sorting like A-buffers Carpenter (1984). These methods introduce time and memory overhead.

To avoid sorting, there are many methods that have been proposed to approximate compositing, namely Order-Independent Transparency (OIT). For instance, $k$-buffer methods work like depth peeling methods, but only storing and accumulating first $k$-layers in a single pass Bavoil et al. (2007). Alternatively, stochastic transparency methods in Monte Carlo rendering samples the fragments according to the opacity and depth, which can generate promising results for a large sampling rate Enderton et al. (2010). A survey on these methods can be found in Wyman (2016).

The weighted blended OIT proposed by McGuire & Bavoil (2013) is the most relevant to our method. There are several OIT variants listed in McGuire & Bavoil (2013), the most general of which is to replace the **OVER** operator with the following commutative blending operator

$$\mathbf{C} = \prod_{i=1}^{\mathcal{N}}(1 - \alpha_i)\,\mathbf{c}_0 + \left(1 - \prod_{i=1}^{\mathcal{N}}(1 - \alpha_i)\right)\frac{\sum_{i=1}^{\mathcal{N}}\mathbf{c}_i\alpha_i w\,(d_i, \alpha_i)}{\sum_{i=1}^{\mathcal{N}}\alpha_i w\,(d_i, \alpha_i)} \tag{2}$$

where $d_i$ is the distance to camera, $\mathbf{c}_0$ is the background color, $w\,(\cdot)$ is a function that decreases with distance, so that objects nearer to the camera are assigned larger weights. In the OIT rendering Eq. 2 there are two sums defining pixel values, and since addition is commutative, they can be computed in any order. Inspired by OIT, our method extends to Gaussian Splatting by introducing the learnable parameters and view-dependent opacity. Besides, compared to the OIT exponential weights, linear weights produce better PSNR results for Gaussian Splatting.

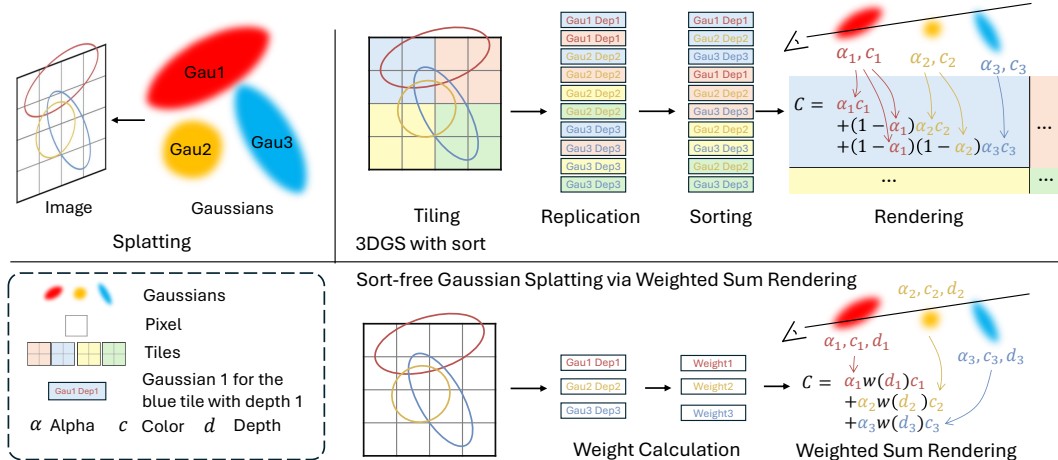

Figure 2: The architecture of sort-free Gaussian Splatting via Weighted Sum Rendering. 3DGS needs to tiling, replication, sorting, and rendering. Our method only needs to calculate the weight for each Gaussian, and independently sum their contributions per-pixel.

## 3  PRELIMINARIES: GAUSSIAN SPLATTING

The 3DGS scene model $\mathcal{G} = \{(\mathbf{p}_i, t_i, \mathbf{q}_i, \mathbf{s}_i, \mathbf{H}_i)\}_{i=1}^{\mathcal{N}}$ represents a scene of $\mathcal{N}$ Gaussians with center locations $\mathbf{p} \in \mathbb{R}^3$, maximum opacity $t_i \in [0, 1]$, orientation $\mathbf{q} \in \mathbb{R}^4$, scale $\mathbf{s} \in \mathbb{R}^3$, and spherical harmonics coefficients $\mathbf{H}$ following an equation similar to Gaussian probability distributions. The opacity of each element at a position $\mathbf{x}$ in 3D space is defined according to

$$\alpha_i(\mathbf{x}) = t_i \exp\left(-\frac{(\mathbf{x} - \mathbf{p}_i)^t \left[\mathbf{\Sigma}(\mathbf{q}_i, \mathbf{s}_i)\right]^{-1}(\mathbf{x} - \mathbf{p}_i)}{2}\right), \quad i = 1, 2, \cdots, \mathcal{N} \tag{3}$$

and considering a camera with focal point at position $\mathbf{f}$, the view-dependent color is defined by

$$\mathbf{c}_i = Y\left(\|\mathbf{f} - \mathbf{p}_i\|, \mathbf{H}_i\right), \quad i = 1, 2, \cdots, \mathcal{N}, \tag{4}$$

where $Y(\cdot)$ indicates the spherical harmonic function and $\|\cdot\|$ indicates the vector norm.

To render an image, each 3D Gaussian is mapped to 2D as approximated by a 2D Gaussian with the following $2 \times 2$ covariance matrix

$$\mathbf{\Sigma}_{\text{2D}} = \mathbf{J}\mathbf{W}\mathbf{\Sigma}_{\text{3D}}\mathbf{W}^t\mathbf{J}^t \tag{5}$$

where $\mathbf{W}$ is a matrix defined by the camera's image-generation transformation, and $\mathbf{J}$ is the Jacobian matrix defined by an affine approximation of the projective camera transformation.

After sorting Gaussians in depth order, the final pixel color obtained by Gaussian splatting can be computed according to

$$\mathbf{C} = \sum_{i=1}^{\mathcal{N}} \mathbf{c}_i \alpha_i \prod_{j=1}^{i-1}(1 - \alpha_j), \tag{6}$$

which corresponds to the well-known computer graphics technique of alpha-blending (Marschner & Shirley, 2015).

## 4  SORT-FREE GAUSSIAN SPLATTING VIA WEIGHTED SUM RENDERING

### 4.1  SORT-FREE GAUSSIAN SPLATTING

As discussed in Sec. 3, 3DGS relies on non-commutative alpha blending, which requires a depth order sorting operation before rendering. To recover high-fidelity of the rendered images, 3DGS models typically include a substantial number of Gaussians, which can escalate to millions in complex scenes. The overhead associated with these sorting operations poses a challenge in splatting Gaussians onto an image without effective optimization.

To accelerate rendering, 3DGS incorporates several optimizations designed to leverage parallel computation capabilities in CUDA. As shown in Figure 2, 3DGS splits the rendered image into multiple non-overlapping tiles, with Gaussians that span across multiple tiles being duplicated accordingly. Each Gaussian is assigned a tile ID and sorted by depth on a per-tile basis. Once sorted, all tiles are rendered in parallel, significantly improving computational efficiency. Additionally, 3DGS employs early termination to further optimize performance: if the alpha value of the frontmost Gaussians is sufficiently high, the rendering process will skip subsequent Gaussians, reducing unnecessary computations and memory usage.

While 3DGS achieved promising results, its CUDA based implementation restricts its portability, and its sorting requirement limits its performance. Their usage of an efficient tile-based radix sort forces them maintain a fully compute-based render pipeline, including compute-based rasterization. On the other hand, current APIs fail to expose efficient mechanisms to submit geometry to the hardware rasterizer in a tiled fashion, thereby requiring that implementations leveraging the hardware rasterizer execute a more costly global sort over all visible Gaussians. 3DGS chose to pursue the fully compute-based approach with its CUDA implementation, which denies them some of the benefits of the GPU's fixed function hardware. Their tile-based sort incurs some overhead in its duplication of Gaussians at tile boundaries, which is exacerbated in instances where there are a large number of Gaussians, and finally, as noted by prior work Radl et al. (2024), sorting Gaussians by their centers can introduce "popping" artifacts during view transformation, further affecting visual quality.

These limitations motivate the elimination of the sorting phase in Gaussian Splatting, which not only significantly simplifies the implementation but also enhances compatibility with hardware graphics pipelines. Inspired by Order Independent Transparency (OIT), we propose a novel approach that modifies the volumetric rendering process in 3DGS to a more efficient representation termed Weighted Sum Rendering (WSR).

## 4.2 WEIGHTED SUM RENDERING

Figure 2 compares the architectures of the original sort-based 3DGS model, with our proposed sort-free Gaussian Splatting framework. Unlike the volumetric rendering approach employed by 3DGS, our method estimates the transparency of each Gaussian solely based on its depth and learnable parameters, thereby eliminating the need for depth ordering. The final image can be rendered using

$$\mathbf{C} = \frac{\mathbf{c}_B w_B + \sum_{i=1}^{\mathcal{N}} \mathbf{c}_i \alpha_i w(d_i)}{w_B + \sum_{i=1}^{\mathcal{N}} \alpha_i w(d_i)}, \qquad (7)$$

where $\mathbf{c}_B$ and $w_B$ indicate the color and learnable weight of the background, respectively. $d$ indicates the depth. $w(\cdot)$ indicates the learnable weight function. Our network learns the Gaussian's parameters during training. Rendering with Eq. 7 in WSR corresponds to computing weighted sums. As the addition is commutative, these

Figure 3: Three variants of Weighted Sum Rendering with different weight calculations, namely Direct Weighted Sum Rendering (DIR-WSR), Exponential Weighted Sum Rendering (EXP-WSR), and Linear Correction Weighted Sum Rendering (LC-WSR).

sums can be computed in any order, overcoming the constraints of depth sorting. Unlike traditional OIT methods Meshkin (2007); McGuire & Bavoil (2013), which rely on predefined parameters, our method optimizes the parameters of this new representation during training.

3DGS produces novel views based on a physics-based blend model, whereas GS-WSR demonstrates the benefits of departing from a phsyically-based model, and using machine learning to train the parameters for a non-physically based model. In volumetric rendering Eq. 6, it is necessary to have $\alpha_i \in [0, 1]$ to guarantee that all terms are positive. However, these constraints are not necessary in WSR Eq. 7 as $\alpha_i$ serves as a learnable parameter in a radiance field model. Removing such constraints can potentially result in better approximations. Similarly, our view-dependent opacity may not correspond to optical laws, but it is in practice useful for minimizing the limitations of the WSR Eq. 7 compared to volume rendering Eq. 6.

**Direct Weighted Sum Rendering (DIR-WSR).** A straightforward method of rendering is to sum all contributions directly, allowing the network to learn specific opacities. In DIR-WSR, the weight

is defined as a constant as follows

$$w(d_i) = 1, \quad i = 1, 2, \cdots, \mathcal{N}. \tag{8}$$

However, DIR-WSR does not work well for complex scenes, and often introduces blurring artifacts in areas where Gaussians overlap. We attribute the poor visual performance due to the lack of depth information, which prevents accurate estimation of transparency. A visual example is provided in Figure 3, illustrating two non-transparent Gaussians with $\alpha = 1$ with different color[1]. Since DIR-WSR uses a constant weight of 1, it is unable to correctly handle this situation, leading to an incorrect estimation of the resulting color as purple.

**Exponential Weighted Sum Rendering (EXP-WSR).** To better capture the effects of occlusion, we introduce EXP-WSR to assign larger weights to Gaussians closer to the camera. The weight is defined as

$$w(d_i) = \exp\left(-\sigma d_i^{\beta}\right), \quad i = 1, 2, \cdots, \mathcal{N}, \tag{9}$$

where $\sigma$, $\beta$ are learnable parameters. In this manner, the Gaussians closer to the viewer would have a higher weight, thus contributing more to the final rendered image. Although EXP-WSR can effectively reduce artifacts and generate better results compared to DIR-WSR, it is not entirely free from visual distortions, as distant Gaussians still contribute to the rendered image to some extent. As shown in Figure 3, the red Gaussian, being closer to the viewer, has a larger weight than the blue Gaussian, yielding results that are closer to the ground truth. However, some artifacts remain visible in the final output.

**Linear Correction Weighted Sum Rendering (LC-WSR).** Inspired by the deformable convolution Dai et al. (2017) and KPConv Thomas et al. (2019), we use linear correction to estimate the weight from depth by

$$w(d_i) = \max\left(0, 1 - \frac{d_i}{\sigma}\right) v_i, \quad i = 1, 2, \cdots, \mathcal{N}, \tag{10}$$

where $\sigma$, $v_i$ are learnable parameters. This formulation will assign a relatively larger weight for the Gaussians closer to the camera. For distant Gaussians, the weight may be reduced to 0, depending on $\sigma$ or $v_i$. As shown in Figure 3, the rendered result is closest to the ground truth, since this model can set weights to zero it can more accurately model occlusions. Additionally, compared to EXP-WSR, LC-WSR is cheaper to compute.

## 4.3 VIEW-DEPENDENT OPACITY

In Sec 4.2, we proposed WSR to assign different weights to Gaussians according to their depth. Moreover, the order of Gaussians would also change depending on the viewer's direction in the original 3DGS. Figure 4 illustrates this phenomenon: two viewers observe two Gaussians from different directions. The left viewer assigns a larger weight to the red Gaussian, while the right viewer assigns a smaller weight to the same Gaussian. This observation motivates our approach to assigning view-dependent opacities.

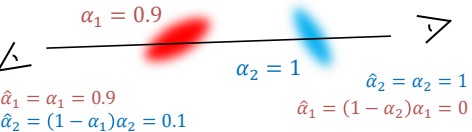

Figure 4: View dependent opacity. In 3DGS, the accumulated $\hat{\alpha}$ changes depending on the viewer's direction, which motivates us to assign view-dependent opacities in sort-free Gaussian Splatting.

We substitute our opacity values for an additional set of spherical harmonic coefficients for view-dependency. It's another mechanism by which we mitigate the contribution of occluded Gaussians. We modify Eq 3, which defines the sort-free Gaussian's opacity, replacing the 3DGS element's maximum opacity $t_i \in [0, 1]$ with an unconstrained value $u_i$ as

$$u_i = Y\left(\|\mathbf{f} - \mathbf{p}_i\|, \mathbf{H}_i\right), \quad i = 1, 2, \cdots, \mathcal{N}, \tag{11}$$

where $Y(\cdot)$ indicates the spherical harmonic function and $\|\cdot\|$ indicates the vector norm. $\mathbf{H}_i$ indicates the learnable spherical harmonics coefficients for opacity. Thus, the maximum opacity $u_i$ depends on view direction $\|\mathbf{f} - \mathbf{p}_i\|$ according to learned spherical harmonics parameter vector $\mathbf{H}_i$. There is a

---

[1]Please note that in WSR, $\alpha$ can be larger than 1. We assume $\alpha = 1$ as non-transparency for illustrative purposes.

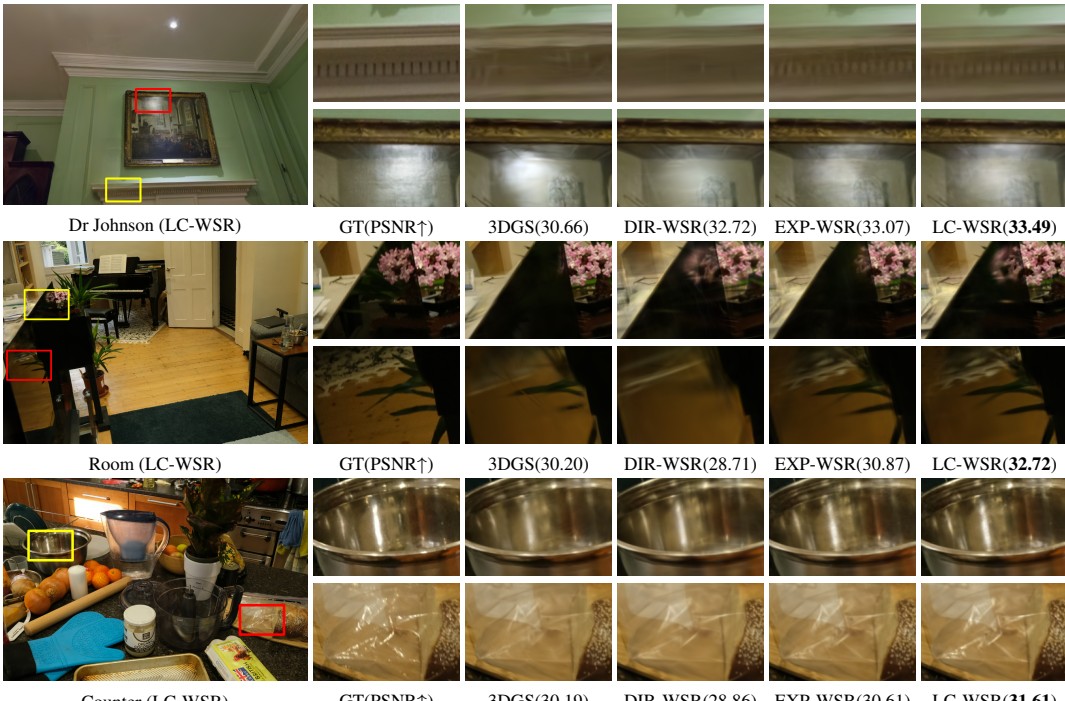

Figure 5: Visual comparison on the Mip-NeRF 360 dataset. Our OIT method achieves similar visual performance compared to 3DGS. Please note that our method doesn't require the order of Gaussians.

slight difference to the SH evaluation for the RGB channels and the opacity in our implementation. The maximum opacity $u_i$ is not clamped. Please note that the view-dependent opacity may not correspond to optical laws, but it is in practice useful for minimizing the limitations of OIT rendering Eq. 7 compared to volume rendering Eq. 6.

### 4.4 IMPLEMENTATION DETAILS

**Loss function**. We optimize our sort-free Gaussian Splatting using the same rendering loss from 3DGS Kerbl et al. (2023), which contains the $\ell_1$ loss and D-SSIM loss with a factor of 0.2.

**Training and Evaluation**. We trained our sort-free Gaussian Splatting method using PyTorch. We implemented the custom CUDA kernels for the PyTorch version. For EXP-WSR, we initialize the $\sigma = 0.1$ and $\beta = 0.8$. For LC-WSR, we initialize $\sigma = 10$ and $v_i = 0.1$. For convenience, we also evaluate the quality of our method with a PyTorch inference simulation.

**Testing on mobile devices**. To measure model efficiency on mobile devices, we implemented our method and the competitive 3DGS method in Vulkan [2], which is a cross-platform graphics and compute API created and maintained by the Khronos Group [3].

3DGS's rendering on mobile device contains four steps: pre-processing, lighting, sorting, and rasterization. The pre-processing step projects the 3D Gaussian into 2D based on the camera view. The lighting stage calculates per-Gaussian color based on the camera view and the spherical harmonics (SHs). The sorting step sorts the Gaussians front to back, and the rasterization step rasterizes all Gaussians within each tile to output the final image.

To make a fair comparison, we implemented two versions of 3DGS, namely 3DGS-Compute and 3DGS-Graphics. 3DGS-Compute attempts to faithfully replicate the approach from 3DGS Kerbl et al. (2023) in Vulkan compute shaders. It follows a similar scheme of replicating Gaussians per-overlapped tile, followed by compute-based rasterization. The most notable difference is that the optimized radix-sort from the NVIDIA CUB libraries NVIDIA (2024) had to be replaced with a Vulkan equivalent. By contrast, the 3DGS-Graphics approach allows for more mobile friendly

---

[2]https://www.vulkan.org
[3]https://www.khronos.org

Table 1: PSNR scores of our method on the Mip-NeRF360 dataset, the Tanks & Temples dataset, and the Deep Blending dataset. Our method achieved comparable results with 3DGS.

| Method | Mip-NeRF 360 | | | Tanks & Temples | | | Deep Blending | | |
|---|---|---|---|---|---|---|---|---|---|
| | PSNR↑ | SSIM↑ | LPIPS↓ | PSNR↑ | SSIM↑ | LPIPS↓ | PSNR↑ | SSIM↑ | LPIPS↓ |
| Plenoxels | 23.08 | 0.626 | 0.463 | 21.08 | 0.719 | 0.379 | 23.06 | 0.795 | 0.510 |
| INGP-Base | 25.30 | 0.671 | 0.371 | 21.72 | 0.723 | 0.330 | 23.62 | 0.797 | 0.423 |
| INGP-Big | 25.59 | 0.699 | 0.331 | 21.92 | 0.745 | 0.305 | 24.96 | 0.817 | 0.390 |
| M-NeRF360 | **27.69** | 0.792 | 0.237 | 22.22 | 0.759 | 0.257 | 29.40 | 0.901 | 0.245 |
| 3DGS | 27.21 | **0.815** | 0.214 | 23.14 | 0.841 | 0.183 | 29.41 | **0.903** | 0.243 |
| Compact 3DGS | 27.08 | 0.798 | 0.247 | **23.71** | **0.845** | 0.178 | **29.79** | 0.901 | 0.258 |
| LightGaussian | 27.28 | 0.805 | 0.243 | 23.11 | 0.817 | 0.231 | - | - | - |
| Compressed 3DGS | 26.98 | 0.801 | 0.238 | 23.32 | 0.832 | 0.194 | 29.38 | 0.898 | 0.253 |
| LC-WSR | 27.19 | 0.804 | **0.211** | 23.61 | 0.842 | **0.177** | 29.63 | 0.902 | **0.229** |

execution. It's implementation globally sorts the Gaussians, eliminating per-tile replication, and allowing it to easily submit a global list of Gaussian to the hardware graphics pipeline. Gaussian color and opacity are evaluated in a fragment shader. These modifications allow Gaussians to be efficiently rendered by the fixed-function rasterizer, while their contributions are accumulated by hardware blending operations.

For both our method and the 3DGS methods, the pre-processing and lighting steps remain the same as the original 3DGS implementation, except that our method's opacity is now calculated in the lighting stage based on the camera view and SHs. Our method removes the entire sorting step, and rasterizes Gaussians in a fragment shader. Besides, our method contains an extra subpss to perform the final normalization step. With our method, all Gaussians are rendered using a single instanced draw call instead of the per-tile fashion used by 3DGS. We verified the consistency between our on mobile implementation versus the Pytorch reference implementation. Please refer to our supplementary for more details about our on mobile implementation.

## 5 EXPERIMENTAL RESULTS

**Datasets**. To ensure a fair comparison, we followed the evaluation setting of 3DGS Kerbl et al. (2023) and conducted our experiments on 13 real-world scenes. Specifically, they include the complete set of scenes from the Mip-Nerf360 dataset Barron et al. (2022), two scenes from the Tanks & Temples dataset Knapitsch et al. (2017), and two scenes from the Deep Blending dataset Hedman et al. (2018). These scenes encompass various challenging scenarios, including both indoor and outdoor environments.

**Evaluation metrics**. We evaluate reconstruction fidelity using Peak Signal-to-Noise Ratio (PSNR), Structural Similarity (SSIM), and Learned Perceptual Image Patch Similarity (LPIPS) metrics. For model efficiency, we report run times on a Qualcomm® Adreno™ GPU from Snapdragon 8 gen 3 chipset. To enable comparison, we also re-implemented our method and the official 3DGS (Kerbl et al., 2023) in Vulkan for this setting.

### 5.1 COMPARISON WITH STATE-OF-THE-ART METHODS

We compare our method with state-of-the-art techniques, including Plenoxels Fridovich-Keil et al. (2022), INGP Müller et al. (2022), M-NeRF360 Barron et al. (2022), 3DGS Kerbl et al. (2023), Compact 3DGS Morgenstern et al. (2023), Compressed 3DGS Niedermayr et al. (2024), and Light-Gaussian Fan et al. (2023). The results in Table 1 demonstrate that our method, while avoiding any sorting, achieves comparable results to the baseline 3DGS. In terms of PSNR, our method outperforms 3DGS by 0.47dB and 0.22dB on the Tanks & Temples and Deep Blending datasets, respectively, while averaging only 0.02dB less on the Mip-NeRF360 dataset. For visual comparison, Figure 5 shows that our LC-WSR excellently recovers challenging details of the fireplace in the "Dr Johnson" scenes. In other scenes, WSR particularly recovers better fidelity in areas with strong illumination and reflection.

### 5.2 COMPUTATIONAL COMPLEXITY STUDY

In this study, we examine the running time and memory consumption of our methods on the mobile devices, using an Snapdragon 8 Gen 3 GPU, and with an implementation based on Vulkan. To

Table 2: Runtime (ms) comparion using a Snapdragon 8 Gen 3 GPU on the Mip-NeRF360, Tanks & Temples and Deep Blending datasets. The images are rendered at a resolution of $1920 \times 1080$. ("*" indicates that mobile resources are exhausted).

| Method | Task | Mip-NeRF360 | | | | | | | | | Tanks&Temples | | Deep Blending | |
|---|---|---|---|---|---|---|---|---|---|---|---|---|---|---|
| | | bicycle | flowers | garden | stump | treehill | room | counter | kitchen | bonsai | truck | train | drjohnson | playroom |
| 3DGS-Compute | Pre-processing | 12.84 | 7.05 | 10.15 | 6.94 | 7.53 | 3.99 | 3.02 | 4.46 | 3.12 | 6.29 | 3.03 | 7.50 | 6.04 |
| | Lighting | 4.57 | 3.26 | 5.19 | 3.43 | 3.37 | 1.84 | 1.55 | 3.07 | 1.40 | 3.53 | 2.41 | 2.61 | 2.40 |
| | Sorting | 246.65 | 122.68 | 210.35 | 180.23 | 139.98 | 70.83 | 50.57 | 89.96 | 50.63 | 105.83 | 60.83 | 140.64 | 110.97 |
| | Rasterization | 1239.12* | 115.18 | 608.19* | 265.27 | 147.08 | 140.16 | 85.94 | 177.01 | 73.43 | 165.64 | 130.23 | 212.10 | 167.09 |
| | Total | 1511.96 | 253.60 | 842.15 | 463.06 | 303.68 | 219.31 | 143.03 | 277.35 | 130.58 | 285.11 | 198.01 | 367.93 | 290.32 |
| 3DGS-Graphics | Pre-processing | 13.27 | 7.82 | 13.73 | 10.46 | 8.01 | 3.40 | 2.62 | 4.09 | 2.62 | 5.84 | 2.35 | 7.22 | 5.57 |
| | Lighting | 5.32 | 3.66 | 5.67 | 3.68 | 3.80 | 2.00 | 1.61 | 3.55 | 1.51 | 4.03 | 2.68 | 2.94 | 2.71 |
| | Sorting | 40.26 | 24.79 | 38.25 | 31.09 | 25.47 | 11.23 | 9.47 | 13.76 | 9.17 | 19.16 | 7.43 | 23.13 | 18.43 |
| | Rasterization | 618.83* | 246.47* | 697.81* | 344.72* | 208.28* | 32.50 | 28.22 | 52.84 | 30.86 | 78.24 | 39.70 | 164.25* | 82.45 |
| | Total | 678.08 | 283.11 | 755.89 | 390.33 | 245.95 | 49.46 | 42.25 | 74.56 | 44.47 | 107.64 | 52.48 | 197.86 | 109.46 |
| Ours | Pre-processing | 6.83 | 3.92 | 5.53 | 6.46 | 4.63 | 1.94 | 1.67 | 2.74 | 1.64 | 3.42 | 1.61 | 4.61 | 2.48 |
| | Lighting | 4.07 | 2.81 | 4.36 | 3.67 | 3.23 | 1.41 | 1.35 | 2.98 | 1.16 | 3.21 | 2.31 | 2.91 | 1.70 |
| | Rasterization | 67.84 | 28.30 | 57.84 | 57.25 | 52.99 | 45.44 | 32.82 | 68.51 | 37.52 | 52.89 | 48.98 | 78.11 | 49.87 |
| | Total | 78.99 | 35.28 | 67.90 | 67.63 | 61.11 | 49.02 | 34.06 | 74.47 | 40.63 | 59.75 | 53.13 | 85.85 | 54.28 |

make a fair comparison, we also re-implement the 3DGS in Vulkan. Please refer Section 4.4 for implementation details. We conduct our experiments on the Mip-Nerf360, Tanks & Temples, and Deep Blending datasets, where each image is rendered at a resolution of $1920 \times 1080$.

Table 2 reports the runtime comparison of 3DGS-Compute, 3DGS-Graphics, and our LC-WSR on the Mip-NeRF360, Tanks&Temples, and Deep Blending datasets. Compared to 3DGS-Compute, 3DGS-Graphics is more efficient as it eliminates the replication steps and thus reducing the number of Gaussians for sorting. Besides 3DGS-Graphics can also better utilize the graphic fixed-functions in the hardware-supported graphic rasterization pipeline. Our method is in most cases faster than the 3DGS-Graphics method due to the elimination of the sorting step and fewer Gaussians. Note that for certain scenes using the 3DGS-Graphics method, we can exhaust our edge device's resources due to the sorting step and high Gaussian count, which will drastically slow down the rasterization step (labeled with "*"). Unfortunately, it can be difficult to pinpoint the sources of inefficiency in complex GPUs. To the best of our knowledge, this is caused by sudden changes to many more cache misses. In contrast, our method is fully capable of running all scenes in real time. In theory, our method's lighting pass should be slightly slower because opacity is calculated using SH, and our rasterization step should be slower due to the weight calculations in the fragment shader and the extra subpass for weight normalization. However, we observed that WSR tends to generate models with fewer Gaussians (scene average 2.88M) compared to the original 3DGS (scene average 3.98M), which is consistent with prior work Radl et al. (2024) that suggests modifying opacity leads to a reduction in the number of Gaussians. This also contributes to faster lighting and rasterization steps for us compared with 3DGS-Graphics. For certain scenes (kitchen and train) that have a similar number of Gaussians, our total time is close to or slightly worse compared to the original model because the overhead added in the rasterization step is not fully compensated by the removal of the sorting step, which may indicate further optimizations are needed for our method's rasterization implementation. Overall, the proposed method is on average 1.23× faster in total time when the mobile resources are not exhausted.

Table 3 compares the runtime memory of our method with 3DGS-Graphics and 3DGS-Compute. Compared to 3DGS-Compute, 3DGS-Graphics utilizes less memory as it performs global sorting thereby eliminating the replication step. Our method only requires around 63% memory of 3DGS-Graphics. This memory reduction can be attributed to removing the sorting operations as well as fewer total Gaussians. Specifically, removing sorting reduces around 17% memory, while the reducing the number of Gaussians contribute the remaining. For the view dependent opacity, the spherical harmonics data in GPU graphic pipeline is represented as RGBARGBA⋯ instead of RBGRGB⋯. We exploit this fact by using the alpha memory position to create view-dependent opacity, which does not increase the amount of memory and vector operations, i.e., computational complexity. For

Table 3: Runtime memory (MB) comparison on the Mip-NeRF360, Tanks & Temples and Deep Blending datasets. The images are rendered at a resolution of $1920 \times 1080$.

| Method | Mip-NeRF360 | | | | | | | | | Tanks & Temples | | Deep Blending | |
|---|---|---|---|---|---|---|---|---|---|---|---|---|---|
| | bicycle | flowers | garden | stump | treehill | room | counter | kitchen | bonsai | truck | train | drjohnson | playroom |
| 3DGS-Compute | 2017 | 1205 | 1920 | 1636 | 1253 | 542 | 421 | 626 | 429 | 850 | 358 | 1131 | 851 |
| 3DGS-Graphics | 1520 | 911 | 1448 | 1235 | 947 | 413 | 322 | 476 | 328 | 644 | 274 | 855 | 645 |
| Ours | 881 | 526 | 681 | 853 | 618 | 276 | 241 | 367 | 251 | 439 | 217 | 615 | 344 |

Table 4: WSR variants on the Mip-NeRF360, Tanks & Temples and Deep Blending datasets.

| Method | Mip-NeRF 360 | | | Tanks&Temples | | | Deep Blending | | |
|---|---|---|---|---|---|---|---|---|---|
| | PSNR↑ | SSIM↑ | LPIPS↓ | PSNR↑ | SSIM↑ | LPIPS↓ | PSNR↑ | SSIM↑ | LPIPS↓ |
| DIR-WSR | 25.99 | 0.778 | 0.262 | 22.80 | 0.821 | 0.218 | 28.85 | 0.899 | 0.254 |
| EXP-WSR | 26.97 | 0.801 | 0.216 | 23.32 | 0.833 | 0.183 | **29.77** | **0.902** | **0.229** |
| LC-WSR | **27.19** | **0.804** | **0.211** | **23.61** | **0.842** | **0.177** | 29.63 | **0.902** | **0.229** |

general storing of the splatting elements, it indeed increases the footprint per Gaussians because the number of spherical harmonics coefficient values increases from 3 to 4.

## 5.3 Ablation study

**Comparison of WSR variants**. We compare the variants of WSR in Table 4 on Mip-NeRF360, Tanks & Temples, and Deep Blending datasets. LC-WSR achieved the best performance on Mip-NeRF360 and Tanks & Temples datasets, while EXP-WSR provides the best performance on Deep Blending dataset. See Figure 5 for visual examples that demonstrate LC-WSR performs best at recovering fine details.

**View dependent opacity**. We examine how our method works with view dependent opacity. Experimentally comparing our method with the view independent opacity method used in 3DGS, we demonstrate view dependent opacity can significantly improve the results, shown in Table 5.

**Popping artifacts.** As a side effect of not relying on sorting, our methods also eliminates "popping" artifacts. Figure 6 shows an experiment in which we slightly rotate the camera, causing the black Gaussian to suddenly "pop" out in 3DGS, generating noticeable temporal artifacts, whereas our method produces temporally consistent results.

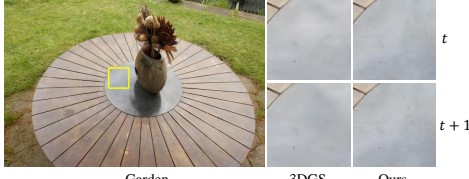

Figure 6: Our method eliminates the "popping" artifacts during view transformation.

**Learnable parameters**. Traditional OIT methods rely on predefined parameters, while our method optimizes the parameters during training. We compare fixed parameters vs. our learnable parameters in Table 5, which shows that our learnable parameters significantly improve the performance.

Table 5: Ablation study on the Mip-NeRF360, Tanks & Temples and Deep Blending datasets.

| Method | Mip-NeRF 360 | | | Tanks & Temples | | | Deep Blending | | |
|---|---|---|---|---|---|---|---|---|---|
| | PSNR↑ | SSIM↑ | LPIPS↓ | PSNR↑ | SSIM↑ | LPIPS↓ | PSNR↑ | SSIM↑ | LPIPS↓ |
| Ours | 27.19 | 0.804 | 0.211 | 23.61 | 0.842 | 0.177 | 29.63 | 0.902 | 0.229 |
| w.o. learnable parameters | 23.19 | 0.711 | 0.318 | 21.55 | 0.788 | 0.245 | 27.92 | 0.893 | 0.260 |
| w.o. view-dependent opacity | 25.88 | 0.784 | 0.251 | 21.83 | 0.798 | 0.237 | 29.27 | 0.902 | 0.249 |

## 6 Conclusions

This paper introduces a novel sort-free Gaussian Splatting method, enabled by its use of learned non-commutative blend weight functions, a technique we term Weighted Sum Rendering. We presented three variants of Weighted Sum Rendering: DIR-WSR, EXP-WSR, and LC-WSR. These variants effectively eliminate the need for the sort operation in 3DGS. Additionally, we developed a view-dependent opacity technique that significantly enhances reconstruction fidelity in sort-free Gaussian Splatting. Our experimental results demonstrate that WSR not only achieves competitive visual performance compared to 3DGS but also operates 1.23 times faster on mobile devices.

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

## A  IMPLEMENTATION DETAILS ON MOBILE PHONES

For the 3DGS-Graphic method, the pre-processing and the lighting steps remain the same as the original 3DGS implementation except for that they are now written in GLSL compute shaders. We changed the rasterization step from a compute based pipeline to traditional graphic pipeline using instanced draw, where each Gaussian treated as an instance and covered by a triangle pair in vertex shader. The fragment shader then colors the pixels within a Gaussian's radius. The alpha blending is handled by the hardware automatically at the rendering back-end. With the graphic pipeline, all Gaussians are rendered using a single instanced draw call instead of the per-tile fashion used by 3DGS. The sorting step is also changed from a tile-based sort to global sort, which is implemented using GLSL compute shader.

For our method, the pre-processing and the lighting stages remain the same as the 3DGS method in Vulkan except for that now opacity is calculated in the lighting stage based on camera view and SHs. Sorting step is removed and we also pass depth for each Gaussian to the graphic pipeline to calculate the per-Gaussian weight. The alpha blending operation is changed from using the blending factor to simple summation of the color and weight. We added an extra graphic pass using the Vulkan subpass feature to handle the final normalization. To make the 3DGS-Graphic method and our method a fair comparison, the render target for both is set to use 16-bit RGBA. Note that the original 3DGS can be implemented using 8-bit RGBA with a slightly lower image quality and better rendering speed, but because our method requires the color and weight to be accumulated, we used 16-bit to avoid overflow issue.

## B  PER SCENES EVALUATION METRIC

Table 6, 7, and 8 present the PSNR, SSIM, and LPIPS metrics for each scene within Mip-NeRF360, Tanks & Temples, and Deep Blending datasets, respectively.

## C  MORE DISCUSSIONS

**Compared with gsplat webgl gsplat.tech** At the same rendering resolution, our method improve the rendering speed from 19-25 fps to 30fps on a Snapdragon 8 gen3. The gsplat implementation runs at a significantly lower resolution and leverages an asynchronous CPU sort running at a frequency below the framerate, which exacerbates temporally artifacts under motion. Our implementation does not have these artifacts, which demonstrates the effectiveness of our method.

**Moving camera along $z$ direction.** The alpha and weights are applied equally to the RGB color components. Ratios of different splats are maintained along the z-direction with Exponential Weighted Sum Rendering (EXP-WSR), but not with Linear Correction Weighted Sum Rendering (LC-WSR). However, LC-WSR also produces visually plausible results. We conducted the experiments to move the camera along $z$ direction. In the experiments, we test our EXP-WSR and

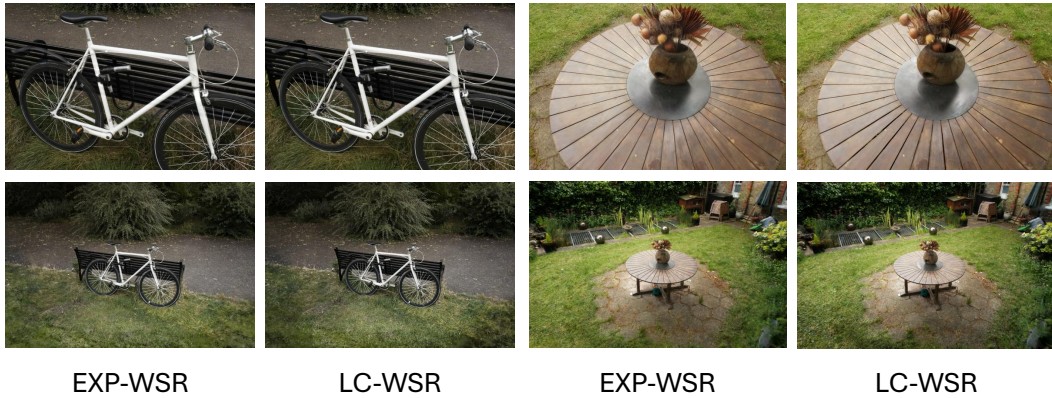

|  EXP-WSR | LC-WSR | EXP-WSR | LC-WSR |

Figure 7: Moving camera along $z$ direction. EXP-WSR and LC-WSR produces plausible results.

LC-WSR on the Bicycle and Garden scenes. As shown in Figure 7, we don't observe the color shifting artifacts.

**Densification and pruning for view dependent opacity.** The 3DGS densification technique was maintained for our method without any modification. And we removed the pruning based on the opacity threshold. There are researches exploring other signals, such as mask Lee et al. (2024), color Bulò et al. (2024), for densification and pruning, and achieving promising results. Our method will benefit from these works in the future.

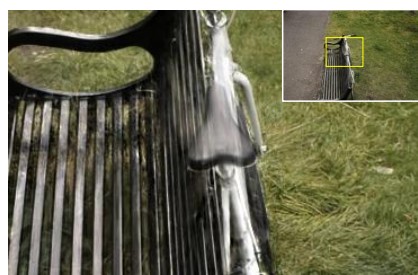

**Culling in WSR.** Culling operates the same as 3DGS. As with 3DGS, we render only those Gaussians within the Camera Frustum. Our implementation does sum over all visible Gaussians, we configure the hardware to perform this summation efficiently using hardware supported blending operations.

Figure 8: Failure case. The seat seems to be transparent, as it is in dark and the bicycle frame is in bright white.

**Failure case.** One type of failure that is easy to identify visually is apparent transparency of dark objects when in front of a very light background. Figure 8 shows an example where our method fails. However, this could be detected with differentiable rendering and thus can be attributed to suboptimal training.

## D    LIMITATIONS AND FUTURE WORK

**Early-Termination**. In 3DGS a pixel can be easily terminated by monitoring if its accumulated alpha has been saturated. Alternately, we could implement heuristic to halt further evaluation of Gaussians whose opacity falls below some threshold, though our current WSR implementation does not support such an optimization. We implemented our method using the traditional graphics pipeline to make efficient use of mobile GPUs, especially the hardware rasterizer. In our implementation, opacity and weight computations are performed in the fragment shader, while blending is handled by the hardware, since every fragment operates independently and does not have access to the accumulated opacity, we are unable to easily implement early termination. This would require reading the render target's content within the invoking draw pass, which necessitates costly read-modify-write operations and requires Vulkan extensions that many mobile devices do not currently support. Additionally, since the screen-space location of each Gaussian is determined only after the vertex shader, a pixel can only be discarded very late in the pipeline.

**Compact Gaussians.** Recently, several compact Gaussian methods have been introduced to significantly reduce the number of Gaussians while maintaining visual performance Jo et al. (2024); Lee et al. (2024); Fan et al. (2023); Niedermayr et al. (2024); Chen et al. (2024); Morgenstern et al. (2023); Navaneet et al. (2023). We believe that our sort-free method can benefit from the rapid pace of research on compact Gaussians.

Table 6: PSNR scores of our method on the Mip-NeRF360 dataset, the Tanks&Temples dataset, and the Deep Blending dataset.

| Method | Mip-NeRF360 | | | | | | | | | | Tanks&Temples | | | Deep Blending | | |
|---|---|---|---|---|---|---|---|---|---|---|---|---|---|---|---|---|
| | bicycle | flowers | garden | stump | treehill | room | counter | kitchen | bonsai | Avg | truck | train | Avg | drjohnson | playroom | Avg |
| Plenoxels | 21.91 | 20.10 | 23.49 | 20.66 | 22.25 | 27.59 | 23.62 | 23.42 | 24.67 | 23.08 | 23.22 | 18.93 | 21.08 | 23.14 | 22.98 | 23.06 |
| INGP-Base | 22.19 | 20.35 | 24.60 | 23.63 | 22.36 | 29.27 | 26.44 | 28.55 | 30.34 | 25.30 | 23.26 | 20.17 | 21.72 | 27.75 | 19.48 | 23.62 |
| INGP-Big | 22.17 | 20.65 | 25.07 | 23.47 | 22.37 | 29.69 | 26.69 | 29.48 | 30.69 | 25.59 | 23.38 | 20.46 | 21.92 | 28.26 | 21.67 | 24.96 |
| Mip-NeRF360 | 24.31 | **21.65** | 26.88 | 26.18 | **22.93** | 31.47 | 29.45 | **31.99** | **33.40** | **27.69** | 24.91 | 19.52 | 22.22 | 29.14 | 29.66 | 29.40 |
| 3DGS | **25.25** | 21.52 | 27.41 | **26.55** | 22.49 | 30.63 | 28.70 | 30.32 | 31.98 | 27.21 | 25.18 | 21.09 | 23.14 | 28.77 | **30.04** | 29.41 |
| Compact 3DGS | 24.77 | 20.89 | 26.81 | 26.46 | 22.65 | 30.88 | 28.71 | 30.48 | 32.08 | 27.08 | 25.35 | 22.07 | 23.71 | 29.06 | 29.87 | 29.46 |
| Compressed 3DGS | 24.97 | 21.15 | 26.75 | 26.29 | 22.26 | 31.14 | 28.67 | 30.26 | 31.35 | 26.98 | 24.82 | 21.86 | 23.34 | 28.87 | 29.89 | 29.38 |
| LightGaussian | 25.20 | 21.54 | 26.96 | 26.77 | 22.69 | 31.40 | 28.48 | 30.87 | 31.41 | 27.13 | **25.40** | 21.84 | 23.44 | - | - | - |
| Ours | 24.20 | 20.45 | **27.78** | 25.39 | 22.01 | **31.93** | **29.53** | 31.38 | 32.05 | 27.19 | 25.28 | **21.93** | **23.61** | **29.25** | 30.00 | **29.63** |

Table 7: SSIM scores of our method on the Mip-NeRF360 dataset, the Tanks&Temples dataset, and the Deep Blending dataset.

| Method | Mip-NeRF360 | | | | | | | | | | Tanks&Temples | | | Deep Blending | | |
|---|---|---|---|---|---|---|---|---|---|---|---|---|---|---|---|---|
| | bicycle | flowers | garden | stump | treehill | room | counter | kitchen | bonsai | Avg | truck | train | Avg | drjohnson | playroom | Avg |
| Plenoxels | 0.496 | 0.431 | 0.606 | 0.523 | 0.509 | 0.8417 | 0.759 | 0.648 | 0.814 | 0.626 | 0.774 | 0.663 | 0.719 | 0.787 | 0.802 | 0.795 |
| INGP-Base | 0.491 | 0.450 | 0.649 | 0.574 | 0.518 | 0.855 | 0.798 | 0.818 | 0.890 | 0.671 | 0.779 | 0.666 | 0.723 | 0.839 | 0.754 | 0.797 |
| INGP-Big | 0.512 | 0.486 | 0.701 | 0.594 | 0.542 | 0.871 | 0.817 | 0.858 | 0.906 | 0.699 | 0.800 | 0.689 | 0.745 | 0.854 | 0.779 | 0.817 |
| Mip-NeRF360 | 0.685 | 0.583 | 0.813 | 0.744 | 0.632 | 0.913 | 0.894 | 0.920 | **0.941** | 0.792 | 0.857 | 0.660 | 0.759 | **0.901** | 0.900 | 0.901 |
| 3DGS | **0.771** | **0.605** | 0.868 | 0.775 | **0.638** | 0.914 | 0.905 | 0.922 | 0.938 | **0.815** | 0.879 | **0.802** | 0.841 | 0.899 | **0.906** | **0.903** |
| Compact 3DGS | 24.77 | 20.89 | 26.81 | 26.46 | 22.65 | 30.88 | 28.71 | 30.48 | 32.08 | 27.08 | 25.35 | 22.07 | 23.71 | 29.06 | 29.87 | 29.46 |
| Compressed 3DGS | 24.97 | 21.15 | 26.75 | 26.29 | 22.26 | 31.14 | 28.67 | 30.26 | 31.35 | 26.98 | 24.82 | 21.86 | 23.34 | 28.87 | 29.89 | 29.38 |
| LightGaussian | 25.20 | 21.54 | 26.96 | 26.77 | 22.69 | 31.40 | 28.48 | 30.87 | 31.41 | 27.13 | **25.40** | 21.84 | 23.44 | - | - | - |
| Ours | 0.744 | 0.580 | **0.872** | 0.728 | 0.614 | **0.925** | 0.909 | **0.923** | 0.938 | 0.804 | **0.882** | 0.802 | **0.842** | 0.898 | **0.906** | 0.902 |

Table 8: LPIPS scores of our method on the Mip-NeRF360 dataset, the Tanks&Temples dataset, and the Deep Blending dataset.

| Method | Mip-NeRF360 | | | | | | | | | | Tanks&Temples | | | Deep Blending | | |
|---|---|---|---|---|---|---|---|---|---|---|---|---|---|---|---|---|
| | bicycle | flowers | garden | stump | treehill | room | counter | kitchen | bonsai | Avg | truck | train | Avg | drjohnson | playroom | Avg |
| Plenoxels | 0.506 | 0.521 | 0.386 | 0.503 | 0.540 | 0.4186 | 0.441 | 0.447 | 0.398 | 0.463 | 0.335 | 0.422 | 0.379 | 0.521 | 0.499 | 0.510 |
| INGP-Base | 0.487 | 0.481 | 0.312 | 0.450 | 0.489 | 0.301 | 0.342 | 0.254 | 0.227 | 0.371 | 0.274 | 0.386 | 0.330 | 0.381 | 0.465 | 0.423 |
| INGP-Big | 0.446 | 0.441 | 0.257 | 0.421 | 0.450 | 0.261 | 0.306 | 0.195 | 0.205 | 0.331 | 0.249 | 0.360 | 0.305 | 0.352 | 0.428 | 0.390 |
| Mip-NeRF360 | 0.301 | 0.344 | 0.170 | 0.261 | 0.339 | 0.211 | 0.204 | 0.127 | 0.176 | 0.237 | 0.159 | 0.354 | 0.257 | 0.237 | 0.252 | 0.245 |
| 3DGS | **0.205** | **0.336** | 0.103 | **0.210** | 0.317 | 0.220 | 0.204 | 0.129 | 0.205 | 0.214 | 0.148 | **0.218** | 0.183 | 0.244 | 0.241 | 0.243 |
| Compact 3DGS | 0.286 | 0.399 | 0.161 | 0.278 | 0.363 | 0.209 | 0.205 | 0.131 | 0.193 | 0.247 | 0.163 | 0.240 | 0.201 | 0.258 | 0.258 | 0.258 |
| Compressed 3DGS | 0.240 | 0.358 | 0.144 | 0.250 | 0.351 | 0.231 | 0.215 | 0.140 | 0.217 | 0.238 | 0.161 | 0.226 | 0.194 | 0.254 | 0.252 | 0.253 |
| LightGaussian | 0.218 | 0.352 | 0.122 | 0.222 | 0.338 | 0.232 | 0.220 | 0.141 | 0.221 | 0.237 | 0.155 | 0.239 | 0.202 | - | - | - |
| Ours | **0.205** | 0.342 | **0.097** | 0.235 | **0.311** | **0.197** | **0.191** | **0.125** | **0.199** | **0.211** | **0.136** | 0.219 | **0.177** | **0.233** | **0.225** | **0.229** |

