# OpenReview forum: "Sort-free Gaussian Splatting via Weighted Sum Rendering"
_ICLR.cc/2025/Conference — ICLR 2025 Poster_

### Official Review · Reviewer_uk93 · 2024-10-26

**Soundness:** 3
**Presentation:** 4
**Contribution:** 4
**Rating:** 8
**Confidence:** 4

**Summary:**

This work attempts to solve a key problem with current Gaussian Splatting techniques, the computational load that occurs during rendering as a result of the non-commutativity of the opacity calculation needed during the rendering process. The authors achieve this by a novel weighting of term added to each Gaussian to replace the traditional opacity term. The weighting term allows for an order independent formulation of the computation of the final colour of the a rendered ray through the field. View-dependence is obtained by the introduction of a spherical harmonic based formulation to the new weighting term similar to that currently used in prior-art Gaussian Splatting methods for view-dependent colour. Performance is evaluated against prior-art techniques.

**Strengths:**

This work solves a crucial problem in real-time rendering of Gaussian Splatting based radiance fields. By removing the need for depth-based sorting of the Gaussians before rendering, critical memory requirements are reduced enabling the implementation of the methodology on mobile devices or other applications with limited computational capacity. So I feel that the work represents a strong contribution to the field. Other strengths include:
1. Clear and easy to read figures and tables clearly detailing the benefits of the proposed method.
2. A detailed and clear explanation of the proposed method. It is clear what has been done, how this is motivated by previous literature.
3. A strong experimental validation. Comparisons are made against well-recognized metrics and a the latest state-of-the-art methods. The experimental results support well the conclusions reached.

**Weaknesses:**

This work is of quite a high quality. I do not see any specific weaknesses in the methodology or the experimental results. There is one specific minor weakness in the experimental section wherein I would like further information on the memory requirements of the proposed method. I detail the exact question I have in the "Questions" section below.
One final minor issue is the presence of minor grammar errors in the text as I mention in the "Questions" section below.

**Questions:**

1. In section 4.3 the authors reveal that view-dependent opacity is obtained by the addition of a new set of spherical harmonics related to the depth weighting introduced by the authors. In the experimental section, the authors show that this doesn't impact memory usage during rendering and a rationale given in the removal of the sorting step and the potential reduction in the number of Gaussians. However, this appears to be referring to instantaneous memory requirements. How does the introduction of the new set of spherical harmonics for each Gaussian affect post-training storage requirements of the radiance field? Is a greater storage footprint needed for the radiance fields produced by the proposed methods?
2. Line 53 "maintain rendering differentiable" should be "maintain rendering differentiability".
3. Line 141 "Chris Wyman provide a good survey on these methods" should be "Chris Wyman provides a good survey on these methods"
4. Line 273, "However, DIR-WSR does not work well for the complex scenes, " should be "However, DIR-WSR does not work well for complex scenes, "
5. Line 307 ends with an additional full stop.

---

> ### Author Response · Authors · 2024-11-24
>
> Thank you for your constructive and thoughtful comments as well as your support, which have significantly enhanced the quality of our submission. Revised part is showed in `teal` in the paper pdf. Please note that we moved the supplementary materials to the end of the main paper. Below, we provide our detailed responses.
>
> > 1. How does the introduction of the new set of spherical harmonics for each Gaussian affect post-training storage requirements of the radiance field? Is a greater storage footprint needed for the radiance fields produced by the proposed methods?
>
> In Table 3, we reported the runtime memory. In the GPU graphic pipeline, the spherical harmonics data needs to be stored as RGBARGBA$\cdots$ instead of RGBRGB$\cdots$, and similarly SIMD instructions operate on vectors of dimension 4. We exploit this fact by using the alpha memory position to create view-dependent opacity, which does not increase the amount of memory and vector operations. For general storing of the splatting elements, it indeed increases the footprint per Gaussians because the number of spherical harmonics coefficient values increases from 3 to 4. We update the Table caption as well as the corresponding part in the experiment section in the revised version.
>
>
> > 2. Minor grammar errors
>
> Thanks for finding those typos, They have all been fixed in the latest version.
>
>
> Finally, we incorporated the above discussions into our revision and will also follow other suggestions to further improve our paper. Thank you!

---

> > ### Comment · Reviewer_uk93 · 2024-11-26
> > **Response to rebuttal**
> >
> > I have considered the authors response to my review and believe that they have addressed my comments. Thank you very much for your consideration.

---

### Official Review · Reviewer_b3oc · 2024-10-28

**Soundness:** 3
**Presentation:** 2
**Contribution:** 3
**Rating:** 8
**Confidence:** 3

**Summary:**

This paper proposes a sort-free variant to the 3DGS algorithm. Because 3DGS is a semi-transparent representation, it requires sorting of all primitives which leads to higher running time, more hardware requirements, and lower compatibility on edge devices. Learning from the traditional sort-free rendering algorithm which replaces the sorting with an estimated weight from depth, this paper proposes to eliminate the sorting process of 3DGS completely.

This paper further proposes a view-dependent opacity to better simulate the occlusion effect. By learning spherical harmonics to represent the opacity, it allows the 3D Gaussians to appear and disappear at different angles, which could further mimic the occlusion effect.

As a result, this paper achieves a slightly higher rendering speed and lower GPU memory footprint, better compatibility on edge devices, and comparable rendering quality than 3DGS.

**Strengths:**

**Motivation**
* The motivation of this paper is very clear. It clearly illustrates the drawbacks of the sorting requirement of 3DGS and the possible benefit of removing it. Sort-free transparent object rendering is also a well-studied direction in traditional rendering. Combining the algorithms from traditional computer graphics with 3DGS is well-motivated.

**Method**
* The proposed method transformed the traditional depth-based weighted rendering to a learnable version to better suit the per-scene optimization used in 3DGS, which provides significant performance improvement.
* The proposed view-dependent opacity is an innovative approach that mimic the occlusion effect. Because this SH parameters are optimized together, it compensates for some performance loss from incorrect occlusion estimation without sorting.

**Experiments**
* The performance on normal desktop GPU demonstrates a small improvement in rendering speed and comparable rendering speed. In fact, it is already remarkable that a sort-free rendering algorithm can match the performance of a sorting-based rendering algorithm in a scene full of semi-transparent primitives.
* The performance improvement on the edge device shows a more significant improvement because of the demanding sorting step.

**Weaknesses:**

** Motivation **
* As shown in the paper, the sorting step is usually considered slow, but actually contributes to only 25% of the total rendering time. The real problem preventing 3DGS based method from matching the rendering speed of traditional methods is the alpha blending step. Although this paper removes the sorting step, it still needs the per-pixel alpha blending of Gaussians of calculated weights. As a result, the speed improvement is rather insignificant.
* Rather than removing the sorting step, many works have proposed to removing the redundant Gaussians directly. These methods seem to provide much more significant speed boost. Of course these two method can be run in parallel, but it still weaken the importance of the proposed method.

** Method **
* The proposed method is largely borrowed from the traditional sort-free methods. I think this is acceptable because of the long history of sort-free rendering research in traditional computer graphics, it does negatively affect the novelty of the proposed method slightly.

** Experiments**
* As mentioned above, the improvement on standard desktop GPU is rather insignificant. Since sort-free rendering is essentially an approximation of the sort-based rendering, it might not be a good idea to choose this method for such a small improvement while there is a possibility of very bad performance on some scenes.
* Continuing from the last point, sort-free approximations are usually inaccurate on small and thin structures. For example, I noticed that the proposed method on the bicycle scene in MipNeRF-360 dataset is somewhat lower than the original 3DGS method. This scene contains very thin and fine structures on the bicycle which could be a better illustration of whether the proposed method suffers from similar artifacts like the traditional methods.
* Despite the significant rendering speed improvement on the edge devices, the paper lacks a detailed explanation of how the edge device running out of resources leads to such a big decrease in rendering speed.

**Questions:**

1. Can the authors provide some qualitative results on the Bicycle scene?
2. Can the authors provide some videos showing the removal of the popping effect?
3. Can the authors show some failure cases of the proposed approximation?

I really want to give a positive rating to this paper, because I know it is very difficult to make a sort-free rendering to achieve the same quality as the sort-based version. However, the marginal speed improvement diminishes the significance of this paper. Furthermore, I hope the authors can demonstrate some failure cases of such approximation methods, as they are quite common in traditional sort-free algorithms. Doing this could give the reader a more comprehensive understanding of the proposed method.

---

> ### Author Response · Authors · 2024-11-24
>
> Thank you for your constructive and thoughtful comments, which have significantly enhanced the quality of our submission. Revised part is showed in `teal` in the paper pdf. Please note that we moved the supplementary materials to the end of the main paper. Below, we provide our detailed responses.
>
> > 1. Sorting contributes to only 25% of the total rendering time, while reducing the redundant Gaussians seem to provide much more significant speed boost.
>
> As the reviewer mentioned, these two methods can be run in parallel. We believe the results may be interesting for graphics specialist considering the fact that **low-power** GPU rendering can only be achieved with increasingly more efficient parallelization. Sorting is intrinsically sequential, and thus greatly complicates the hardware parallelization by requiring techniques like tiling, and to remove the ``popping'' requires significantly more complicated sorting. Weighted sums, on the other hand, eliminates dependencies and works for these low-power platforms.
>
>
>
> > 2. The proposed method is largely borrowed from the traditional sort-free methods, which negatively affects the novelty of the proposed method slightly
>
> This paper shows that the OIT idea can be similarly greatly extended. Instead of having a combination of opaque and semi-transparent elements in fixed settings, all 3DGS elements have intrinsic transparency and all parameters are learned according to rendering results, enabling new ideas like learned view-dependent opacity. Another interesting and unexpected finding is that with machine learning it was possible to find that linear weights (LC-WSR), which are less intuitive than OIT exponential weights, not only work well, but in fact produce the best PSNR results.
>
>
> > 3. Explain how the edge device running out of resources leads to such a big decrease in rendering speed.
>
> Our implementation of 3DGS appears to be hitting some sort of GPU bottleneck when rendering scenes with a large number of sorted Gaussians. Unfortunately it can be difficult to pinpoint the sources of inefficiency in complex GPUs. To the best of our knowledge, this is caused by sudden changes to many more cache misses.  We suspect that this bottleneck could be addressed with in-depth analysis and GPU specific optimizations, however, this would make our reference implementation less faithful to the original 3DGS implementation. We note that our method appears to be avoid this bottleneck in its simplicity. We added a comment about this to the paper.
>
>
> > 4. Sort-free approximations are usually inaccurate on small and thin structures. Can the authors provide some qualitative results on the Bicycle scene?
>
> We conducted the suggested experiment to provide additional rendering results on the challenging Bicycle scene, which contains many small and thin structures. The results demonstrate that our method still achieves plausible outcomes in these areas, such as the bicycle spokes. The video can be accessed at: https://anonymous20241121.github.io/sort_free_gs/bicycle.mp4
>
>
> > 5. Can the authors provide some videos showing the removal of the popping effect?
>
> We created a video featuring the well-known "Garden" scene, where the "popping" artifacts are most noticeable. Compared to 3DGS, our method successfully eliminates these artifacts by removing the sorting step. The video can be accessed at: https://anonymous20241121.github.io/sort_free_gs/pop.mp4
>
>
> > 6. Can the authors show some failure cases of the proposed approximation?
>
> One type of failure that is easily identifiable visually is the apparent transparency of dark objects when placed in front of a very light background. However, this issue could be detected with differentiable rendering and can thus be attributed to suboptimal training. In the supplementary material, we have provided a visual example and included a discussion of these failure cases.
>
>
> Finally, we incorporated the above discussions into our revision and will also follow other suggestions to further improve our paper. We hope the shared videos can show that the proposed approach has some limitations, but at the same time view are surprisingly good considering its simplicity. We also want to show that although 3DGS sorting may seem inevitable, machine learning can enable it to be replaced. Thank you!

---

> > ### Comment · Reviewer_b3oc · 2024-11-25
> >
> > I appreciate the detailed response from the authors. I find most of the explanation provided by the author to my doubts about performance and edge cases acceptable. The performance on the challenging Bicycle scene is especially persuasive.
> >
> > However, I still believe that the improvement in rendering speed is not the most important contribution of this paper given the already very high rendering speed of the vanilla 3DGS. I think the simplicity of the proposed method without sorting can be very useful for highly complicated scenarios, such as city-scale reconstruction, streaming, level of details, parallel rendering, etc.
> >
> > In general, my questions are well answered and I appreciate the innovative contribution of this paper to the community. I would like to raise my rating as well.

---

### Official Review · Reviewer_Man9 · 2024-11-01

**Soundness:** 2
**Presentation:** 3
**Contribution:** 3
**Rating:** 6
**Confidence:** 3

**Summary:**

The authors propose a novel rasterization backend for 3D Gaussian splatting (3DGS) that addresses the computational and memory overhead associated with depth sorting. Their primary contribution is replacing traditional alpha blending with a weighted sum rendering approach. The rendering is achieved through direct weighted summation of Gaussians, where weights are determined by a parameterized truncated linear function applied to the product of color and opacity. Additionally, they introduce view-dependent optimizable opacity for individual Gaussians, modeled using spherical harmonics.
Their weighted sum rendering (WSR) technique demonstrates comparable rendering quality to conventional 3DGS. To validate the advantages of their sort-free approach, the authors implement both vanilla 3DGS and WSR on the Vulkan API and evaluate performance on mobile GPU hardware, achieving a 1.23x speedup.

**Strengths:**

- The paper aims to address a fundamental bottleneck in 3DGS rendering by proposing a sort-free pipeline, distinguishing itself from previous approaches that primarily focused on reducing Gaussian counts.
- The investigation of various weight derivation functions (direct sum-up, exponential weighted, and linear correction weighted) demonstrates a relative comprehensive experimental analysis effort.
- Implementation and evaluation on mobile devices effectively demonstrates the practical benefits of the sort-free approach in resource-constrained environments.
- The method appears to mitigate "pop" artifacts, a persistent challenge in volumetric rendering, providing an additional benefit beyond computational efficiency.

**Weaknesses:**

- The theoretical foundation for replacing alpha blending with an algebraic function requires stronger justification. While the proposed representation achieves good results, it lacks the physical insights inherent in traditional alpha blending, which models light obstruction during propagation. The introduction of view-dependent opacity appears necessary for acceptable rendering quality, but the paper would benefit from a more rigorous analysis of why this combination works, rather than empirical validation alone.
- The preference for linear weights over exponential weights raises questions about robustness. When considering camera movement along viewing rays, exponential weights (with β ≈ 1) should theoretically provide more consistent mixing ratios between Gaussians. Linear weights may introduce color shift artifacts due to distance-dependent mixing ratio variations. While the authors claims "some artifacts remain visible" then selected linear weight rather than exponential weight, these issues warrant more detailed examination.
- The method's generalizability is limited by the transformation of opacity into a view-dependent "black box" variable. This fundamentally alters the role of opacity, which traditionally serves as a crucial signal for pruning and densification in many 3DGS applications, potentially limiting compatibility with existing and future research in this domain.

**Questions:**

- Does the method exhibit color shift artifacts during camera movement along the z-direction, as suggested by the concerns regarding linear weighting?
- Since opacity is not capped at 1, how did you enforce the data range of final color, simply clamp?
- How do other Gaussian attributes maintain their geometric meaning in this framework? Specifically: Does the mean of Gaussians still correspond to point cloud locations? How about depth extraction?
- While traditional 3DGS implementations employ culling to manage computational complexity, does this approach require weighted summation of ALL Gaussians? What are the implications for scalability?
- Please clarify the memory usage analysis: does it encompass only Gaussian attributes or include runtime memory requirements?
What is the memory overhead of spherical harmonics coefficients for view-dependent opacity compared to vanilla 3DGS?
- Regarding Table 2, could you elaborate on the starred data points showing significantly higher values? How does resource exhaustion due to sorting impact subsequent pipeline stages?
- Given the potential for hardware-accelerated sorting methods specifically designed for 3DGS, possibly through hardware-software co-design, how might this work adapt to or complement such future developments?

---

> ### Author Response · Authors · 2024-11-24
> **Response to reviewer (1/2)**
>
> Thank you for your constructive and thoughtful comments, which have significantly enhanced the quality of our submission. Revised part is showed in `teal` in the paper pdf. Please note that we moved the supplementary materials to the end of the main paper. Below, we provide our detailed responses.
>
> > 1. The theoretical foundation of the algebraic rendering function requires stronger justification.
>
> Our paper is inspired by the Order Independent Transparency, which is a traditional rendering method. Our method can be considered as an extension of weighted order-independent blending[1] on Gaussian Splatting.  Our main objective was not to provide more insight, but to study how mathematical formulations that enable faster computations can effectively substitute physically-based representations. From this perspective, the addition of view-dependent opacity adds degrees of freedom to enable better solution, but it was motivated by the intuition that WSR needs better control over transparency. Following your suggestion, we have updated the related work section to establish stronger connections between our work and previous rendering methods.
>
> ```
> [1] McGuire, Morgan, and Louis Bavoil. "Weighted blended order-independent transparency." Journal of Computer Graphics Techniques 2.4 (2013).
> ```
>
>
>
> > 2. Moving camera along $z$-direction might introduce the color-shifting artifacts
>
> The alpha and weights are applied equally to the RGB color components. As shown in Eq 7 in the paper, our method includes a final normalization stage. We agree with reviewer that ratios of different splats are maintained along the $z$-direction with Exponential Weighted Sum Rendering (EXP-WSR), but not with Linear Correction Weighted Sum Rendering (LC-WSR). However, LC-WSR also produces visually plausible results. We conducted the suggested experiments by moving the camera along the $z$-direction and tested our EXP-WSR and LC-WSR on the Bicycle and Garden scenes. In the video, we don't observe the color shifting artifacts. We have included a discussion of these findings in the revised paper. The video can be accessed at: https://anonymous20241121.github.io/sort_free_gs/z_dir.mp4
>
>
>
>
> > 3. View dependent opacity potentially limits the compatibility with research in pruning and densification.
>
> The 3DGS densification technique was maintained for our method without any modification, while pruning needs to be extended. We agree with the reviewer that view-dependent opacity is not compatible with some recent densification and pruning methods. However, there are researches exploring other signals, such as mask[2], color[3], for densification and pruning, , which have shown promising results. Our method will benefit from these works in the future.We have included a discussion of this in the revised paper.
>
> ```
> [2] Lee, Joo Chan, et al. "Compact 3d gaussian representation for radiance field." Proceedings of the IEEE/CVF Conference on Computer Vision and Pattern Recognition. 2024.
> [3] Bulò, Samuel Rota, Lorenzo Porzi, and Peter Kontschieder. "Revising densification in gaussian splatting." arXiv preprint arXiv:2404.06109 (2024).
> ```
>
>
> > 4. Since opacity is not capped at 1, how did you enforce the data range of final color, simply clamp?
>
> Weighted Sum Rendering (WSR) is defined as a quotient, with alpha and weight contributions appearing in the numerator and denominator, followed by a final normalization stage, as shown in Eq 7 in the paper. Additionally, our final rendered image is clamped in accordance with the 3D Gaussian Splatting (3DGS) method.
>
>
> > 5. How do other Gaussian attributes maintain their geometric meaning in this framework? Specifically: Does the mean of Gaussians still correspond to point cloud locations? How about depth extraction?
>
> Other Gaussian attributes remain the same as in 3DGS. The $xyz$ coordinates still indicate the location of the Gaussians. In our method, we extract the same depth from 3DGS, which is used for sorting in 3DGS.

---

> > ### Author Response · Authors · 2024-11-24
> > **Response to reviewer (2/2)**
> >
> > > 6. What about the culling in WSR?
> >
> > Culling operates the same as 3DGS. As with 3DGS, we render only those Gaussians within the Camera Frustum. Our implementation does sum over all visible Gaussians, we configure the hardware to perform this summation efficiently using hardware supported blending operations. We added a discussion in the revised paper.
> >
> >
> > > 7. Clarify the memory usage analysis.
> >
> > We compare the runtime memory in Table 3. For view-dependent opacity, the spherical harmonics data in the GPU graphics pipeline is represented as RGBARGBA... instead of RBGRGB... . We exploit this by using the alpha memory position to create view-dependent opacity, which does not increase the runtime memory. However, our model requires more storage due to the additional spherical harmonics coefficients for view-dependent opacity. We have updated the caption of Table 3 and the corresponding section in the experiment part of the paper.
> >
> >
> > > 8. Why does resource exhaustion lead to significantly higher values?
> >
> > Our implementation of 3DGS appears to be hitting some sort of GPU bottleneck when rendering scenes with a large number of sorted Gaussians. Unfortunately, it can be difficult to pinpoint the sources of inefficiency in complex GPUs. To the best of our knowledge, this is caused by sudden changes to many more cache misses.  We suspect that this bottleneck could be addressed with in-depth analysis and GPU specific optimizations, however, this would make our reference implementation less faithful to the original 3DGS implementation. We note that our method appears to avoid this bottleneck in its simplicity. We added a comment about this to the paper.
> >
> >
> > > 9. Given the potential for hardware-accelerated sorting methods specifically designed for 3DGS, possibly through hardware-software co-design, how might this work adapt to or complement such future developments?
> >
> > A significant advantage of our technique is its independence from the latest hardware, making it portable to older, lower-power devices and implementable in legacy APIs such as WebGL. The typical time delay from chip design to market release is approximately three years. Given that Gaussian Splatting is a relatively new technique, only one year old and still undergoing extensive research and development, it is expected that future hardware, particularly state-of-the-art and task-specific 3DGS hardware, will surpass our technique in various aspects. However, our method's compatibility with existing, less advanced hardware remains a notable strength.
> >
> >
> > Finally, we incorporated the above discussions into our revision and will also follow other suggestions to further improve our paper. Thank you!

---

> > > ### Comment · Reviewer_Man9 · 2024-11-25
> > >
> > > Thx for the reply and extensive experiment as well as videos. Though most other reviewers think it is okay for the idea of order-independent blending and your result does show a promising potential for its application on 3DGS, yet I still believe **research is not something "we apply previous idea A on another idea B get C and you see, IT WORKS"**. Not to say the great amount of newly introduced parameters in your proposed method.
> > >
> > > I will keep my marginal accept decision.

---

### Official Review · Reviewer_FZe3 · 2024-11-02

**Soundness:** 3
**Presentation:** 3
**Contribution:** 3
**Rating:** 6
**Confidence:** 4

**Summary:**

This paper proposed a sort-free Gaussian Splatting pipeline, which reduces the computational overhead of rendering to adapt to scenarios with limited computing resources. The key insight of this paper is that the view dependent sorting in the vanilla 3DGS involves a lot of computational overhead and is difficult to port to mobile devices. Therefore, the authors proposed to use weighted sum and view-dependent opacity to approximate alpha-blending. Experiments demonstrate that proposed method achieves comparable rendering quality and faster rendering speed than the original 3DGS on mobile devices.

**Strengths:**

- The author provides a clear analysis of the tile-based sort in the vanilla 3DGS, explains its impact on rendering speed, and explains why it is difficult to implement on other devices.
- The proposed method speeds up rendering while ensuring rendering quality, and has been verified on consumer mobile devices.
- The proposed weighted sum approximation alleviates the *pop* phenomenon caused by inconsistent sorting results under different view-directions in the vanilla 3DGS.

**Weaknesses:**

- The proposed method gets rid of the costly sorting, but a more complex weight and view-dependent opacity are used. Although the author points out in the paper that a general scheme for learning scene representations is bound only by the mathematical constraints of the scene model. According to the experimental results in the paper, the use of simple weighted-sum and view-independent opacity cannot achieve satisfactory rendering results. Does this mean that for rendering processes that are not physically and optically, more complex models must be used, such as introducing more learnable parameters as in the paper?
- In the conclusion part of the paper, the author mentioned some recent research on compact Gaussian Splatting. According to my understanding of these works, since the methods proposed in these works obtain more compact 3D Gaussian representations, they can reduce the sorting overhead and speed up rendering to a certain extent. Therefore, I think the author should add a comparison of rendering quality, speed and memory usage between the proposed method and these works, such as Compact 3DGS[1], LightGaussian[2] and Compressed 3DGS[3] in the experimental evaluation section.

[1] Compact 3D Gaussian Representation for Radiance Field https://openaccess.thecvf.com/content/CVPR2024/papers/Lee_Compact_3D_Gaussian_Representation_for_Radiance_Field_CVPR_2024_paper.pdf

[2] LightGaussian: Unbounded 3D Gaussian Compression with 15x Reduction and 200+ FPS https://arxiv.org/pdf/2311.17245

[3]  Compressed 3D Gaussian Splatting for Accelerated Novel View Synthesis https://openaccess.thecvf.com/content/CVPR2024/papers/Niedermayr_Compressed_3D_Gaussian_Splatting_for_Accelerated_Novel_View_Synthesis_CVPR_2024_paper.pdf

**Questions:**

In the paper, the author points out that for some complex scenes, since a large number of 3D Gaussians are needed for representation, the sorting process will exhaust the resources of mobile devices, making real-time rendering impossible, such as the bicycle scene in Mip-NeRF 360 dataset. However, I tried to visualize the bicycle scene using a gsplat-based webgl viewer on my phone (iPhone 15 Pro Max), and was surprised to find that it achieves real-time rendering, about 20-30fps. This seems to indicate that some optimizations to the 3DGS rendering pipeline (gsplat uses some tricks) can increase the rendering speed on mobile devices without more complex designs. I would like to know the author's opinion on my point of view and more analyses of the key differences that enable real-time performance on mobile devices. And they can test the performance of gsplat on Snapdragon® 8gen3 chipset and compare with their proposed method.

Here is the link of gsplat webgl-viewer https://gsplat.tech

---

> ### Author Response · Authors · 2024-11-24
>
> Thank you for your constructive and thoughtful comments, which have significantly enhanced the quality of our submission. Revised part is showed in `teal` in the paper pdf. Please note that we moved the supplementary materials to the end of the main paper. Below, we provide our detailed responses.
>
>
>
> > 1. Rendering rendering processes that are not physically and optically, more complex models must be used, such as introducing more learnable parameters?
>
>
> We recognize that the term "complexity" can be interpreted in two distinct ways in this context. Our proposal introduces additional spherical harmonic parameters, which may appear more "complex" or complicated from a human perspective. However, our method actually reduces computational complexity and requires less memory compared to 3D Gaussian Splatting (3DGS), as demonstrated in Tables 2 and 3 of our paper. First, our method eliminates the sorting step. Second, the data in GPU graphic pipeline is represented as RGBARGBA... instead of RBGRGB... We exploit this fact by using the alpha memory position to create view-dependent opacity, which does not increase the amount of memory and vector operations, i.e., computational complexity.
>
>
> > 2. Compare with Compact 3DGS, LightGaussian, and Compressed 3DGS
>
>
> Thank you for your suggestion. We updated Table 1 as well as the experiment section to compare rendering quality with Compact 3DGS, LightGaussian, and Compressed 3DGS. We also updated the Tables in the supplementary materials to include the results for each scene. For speed and memory comparisons, we evaluated our method on mobile devices. However, the aforementioned methods are implemented using CUDA, and many features, such as codebooks, are not supported in the mobile GPU graphics pipeline. Consequently, we were unable to perform a direct comparison within the limited time.
>
> Below the updated Table 1. Please check our supplementary materials for results of each scene.
>
> **Table 1: PSNR scores of our method on the Mip-NeRF360 dataset, the Tanks & Temples dataset, and the Deep Blending dataset. Our method achieved comparable results with 3DGS.**
>
>
> | Method         | (M-NeRF360) PSNR↑ | (M-NeRF360) SSIM↑ |(M-NeRF360) LPIPS↓ | (T&T) PSNR↑ | (T&T) SSIM↑ | (T&T) LPIPS↓ | (DB) PSNR↑ | (DB) SSIM↑ | (DB) LPIPS↓ |
> |----------------|-------|-------|--------|-------|-------|--------|-------|-------|--------|
> |                | PSNR↑ | SSIM↑ | LPIPS↓ | PSNR↑ | SSIM↑ | LPIPS↓ | PSNR↑ | SSIM↑ | LPIPS↓ |
> | Plenoxels      | 23.08 | 0.626 | 0.463  | 21.08 | 0.719 | 0.379  | 23.06 | 0.795 | 0.510  |
> | INGP-Base      | 25.30 | 0.671 | 0.371  | 21.72 | 0.723 | 0.330  | 23.62 | 0.797 | 0.423  |
> | INGP-Big       | 25.59 | 0.699 | 0.331  | 21.92 | 0.745 | 0.305  | 24.96 | 0.817 | 0.390  |
> | M-NeRF360      | **27.69** | 0.792 | 0.237  | 22.22 | 0.759 | 0.257  | 29.40 | 0.901 | 0.245  |
> | 3DGS           | 27.21 | **0.815** | 0.214  | 23.14 | 0.841 | 0.183  | 29.41 | **0.903** | 0.243  |
> | Compact 3DGS   | 27.08 | 0.798 | 0.247  | **23.71** | 0.831 | 0.178  | **29.79** | 0.901 | 0.258  |
> | LightGaussian  | 27.28 | 0.805 | 0.243  | 23.11 | 0.817 | 0.231  | -     | -     | -      |
> | Compressed 3DGS| 26.98 | 0.801 | 0.238  | 23.32 | 0.832 | 0.194  | 29.38 | 0.898 | 0.253  |
> | LC-WSR         | 27.19 | 0.804 | **0.211** | 23.61 | **0.842** | **0.177** | 29.63 | 0.902 | **0.229** |
>
>
> > 3. Compared with gssplat webgl https://gsplat.tech
>
> Thank you for highlighting this implementation. We conducted the suggested experiment and observed that, at the same rendering resolution, our method improves the rendering speed from 19-25 fps to 30 fps on a Snapdragon 8 Gen 3. The gsplat implementation operates at a significantly lower resolution and utilizes an asynchronous CPU sort running at a frequency below the framerate, which exacerbates temporal artifacts during motion. Our implementation does not exhibit these artifacts. We have included a discussion of these findings in the revised paper.
>
> Furthermore, as noted at the bottom of gsplat.tech, the author acknowledges that sorting is a bottleneck:
>
>
> ```
> There's artifacts when I turn the model quickly.
>
> WebGL can't do efficient GPU sorting of the splats, so I had to pull some tricks to make it work. The splats are sorted on the CPU and the ordering may take a few frames to get updated on the GPU.
> ```
>
>
> Finally, we incorporated the above discussions into our revision and will also follow other suggestions to further improve our paper. Thank you!

---

> ### Comment · Reviewer_FZe3 · 2024-11-25
> **Official Comment by Reviewer FZe3**
>
> I appreciate the authors for providing such a detailed explanation in response to my questions. And the authors' response have already address all of my concerns. Therefore, I will raise my score. Thanks again to the authors for their efforts during the rebuttal period.

---

### Author Response · Authors · 2024-11-24
**Summary of reviews**

We thank the reviewers for their constructive and thoughtful comments, which have significantly enhanced the quality of our submission. Revised part is showed in `teal` in the paper pdf. Please note that we moved the supplementary materials to the end of the main paper. Below, we provide a general summary of the common strengths and weaknesses identified by the reviewers, followed by specific responses to each reviewer's remarks.


### Strengths:
- The paper is well-motivated, addressing a critical bottleneck in 3D Gaussian Splatting (3DGS) rendering by introducing a sort-free pipeline.
- The proposed learnable weighted sum rendering technique, as well as view-dependent opacity, offer substantial performance improvements.
- The performance gains are particularly notable on mobile devices.
- The method effectively mitigates "pop" artifacts, a persistent challenge in volumetric rendering.

### Weaknesses:
- The paper requires a comparison with methods that reduce the number of Gaussians and with 3DGS implementations in WebGL.
- The analysis of memory usage and GPU resource exhaustion needs clarification.


As a general comment, in this paper we consider the fact that for Gaussian Splatting to be successfully used on mobile devices, it should not only be rendered in real time, but also with low power usage. While many works aim to reduce complexity, our focus is specifically on addressing the sorting problem.

---

### Meta-Review · Area_Chair_2SRi · 2024-12-20

**Metareview:**

The paper presents a novel sort-free rendering pipeline for 3D Gaussian Splatting (3DGS). Inspired by order-independent transparency, the method approximates alpha blending using weighted sums, thereby eliminating the need for sorting and reducing rendering time, particularly on mobile devices. Additionally, the proposed approach mitigates pop artifacts typically caused by sorting.

The motivation is clear, the idea is novel, and the experiments are compelling. The primary weakness of the initial submission, its lack of comparisons with compact 3DGS representations, was effectively addressed in the rebuttal.

Overall, the paper makes a valuable contribution by presenting a novel sort-free rendering pipeline for 3DGS. The implementation and evaluation on mobile devices convincingly demonstrate the practical advantages of this approach in resource-constrained environments.

**Additional Comments On Reviewer Discussion:**

The paper mainly received positive feedback in the initial reviews, with some issues addressed in the rebuttal.

A primary concern was the lack of comparison with compact 3DGS representations, which also aim to improve rendering speed. Additionally, reviewers suggested including comparisons with popular WebGL-based renderers on mobile devices. The revision addressed these concerns by incorporating comparisons with Compact 3DGS, LightGaussian, and Compressed 3DGS. For WebGL-based renderers, the rebuttal compared the proposed method with gssplat and reported performance gains.

Some reviewers questioned the novelty of the method, as it is inspired by traditional sort-free techniques. However, the rebuttal clarified that significant extensions were required to adapt and make the approach work effectively for 3DGS, establishing sufficient novelty.

The rebuttal effectively addressed most of the concerns raised during the review process. All reviewers were optimistic about the paper by the end of the discussion stage.

---

### Decision · Program_Chairs · 2025-01-22

Accept (Poster)